# Ecological Efficiency Measurement and Technical Heterogeneity Analysis in China: A Two-Stage Three-Level Meta-Frontier Network Model Based on Segmented Projection

**Ruiyue Lin** ***, Xinyuan Wang and Yu Jiang**

College of Mathematics and Physics, Wenzhou University, Wenzhou 325035, China; wxy_sylviaff@163.com (X.W.);
22461027005@stu.wzu.edu.cn (Y.J.)
* Correspondence: rachel@wzu.edu.cn

**Abstract:** Due to persistent technological impacts on ecological efficiency (eco-efficiency) and variations in economic power and resource endowments among regions, considering regional and temporal heterogeneity becomes imperative. Ecosystems, often divided into economic production and environmental governance stages, necessitate a holistic assessment incorporating regional, temporal heterogeneity and stage distinctions. To address potential issues of a technology gap ratio (TGR) exceeding 1 within a two-stage network structure with dual heterogeneity, we introduce a segmented projection three-layer meta-frontier analysis method. In empirical study, we systematically examined eco-efficiency, emissions inefficiency and technology gaps across management, regional and temporal dimensions in 30 Chinese provinces from 2016 to 2020. Findings reveal disparities in management eco-efficiency, with the central provinces outperforming the east. Regional differences indicate advanced technology in the east, contributing to superior eco-efficiency. Temporal analysis highlights the positive role of scientific and technological development. Emissions inefficiency improvements are noted, necessitating attention toward management and regional technology levels. Eastern provinces exhibit superior emissions efficiency, emphasizing the role of regional and technological development. Recommendations include prioritizing environmental governance, strengthening regional collaborations and implementing policies to bridge technology gaps.

**Keywords:** data envelopment analysis; eco-efficiency; emissions efficiency; network SBM; meta-frontier; heterogeneity



## 1. Introduction

The concept of ecological efficiency (eco-efficiency) was first introduced by Schaltegger and Sturm [1] and was originally expressed as the fewest ecological resources needed to meet the most human needs [2]. Later, eco-efficiency was defined formally by the World Business Council for Sustainable Development [3]. It includes three key targets: minimizing the use of ecological resources, maximizing the production of goods and services for humanity and minimizing the adverse impact on the environment. Energy efficiency [4], environmental efficiency [5,6], carbon emission efficiency [7] and so on, which are regarded as the subcategories of eco-efficiency, have also attracted wide attention. By evaluating eco-efficiency, sustainable development level can be comprehensively evaluated from the perspectives of resources, the economy and the environment. In view of this, eco-efficiency evaluation has attracted the attention of researchers.

Data envelopment analysis (DEA) [8] is a widely used technique to measure the efficiencies of decision making units (DMUs). It is unit-invariant and can comprehensively consider multiple inputs and outputs; especially, it can deal with bad pollution outputs well. Due to these advantages, eco-efficiency has been widely evaluated by using DEA models in many studies. Many different types of DEA models, such as the CCR [8], BCC [9], directional distance function [10] and slacks-based measure (SBM) model [11] are used

to measure eco-efficiency from various perspectives. Table 1 summarizes the indicators of these DEA studies mentioned above. We find that most of them consider population or industry employees, capital and energy as inputs and GDP or other economic benefit indexes as the output. Most of the undesirable outputs contain $CO_2$ and/or $SO_2$ emissions.

**Table 1.** Indicators of relevant DEA studies.

| | Inputs | Outputs | Undesirable Outputs |
|---|---|---|---|
| Song et al. [12] | Capital, employment | GDP | $SO_2$ |
| Guo et al. [13] | Area, population, energy, energy stock (carry-over) | GDP | $CO_2$ |
| Cheng et al. [14] | Energy, employees, capital | GDP | Waste water, $SO_2$ |
| Xie et al. [15] | $CO_2$, Energy, population | GDP | – |
| Shang et al. [16] | Energy, capital, labor | GDP | $CO_2$, $SO_2$ |
| Chen and Lin [17] | Energy, capital, labor | Gross industrial output | $CO_2$ |
| de Araújo et al. [18] | Energy, water, population, vehicles, territorial area, felled area | Ratio of gross domestic, product per capita to area deforested | – |
| Teng et al. [19] | Population, energy, capital | GDP | $CO_2$, afforestation area |
| Zheng [20] | Employment, capital, electricity | GDP | $SO_2$, soot, waste water, PM2.5, Energy |
| Matsumoto and Chen [21] | Energy, capital, employees, water | Industrial added value | Waste gas, waste water, solid waste, $CO_2$ |

If there are obvious differences in environmental characteristics, e.g., economic bases and resource endowment, measuring all DMUs at a similar production technological level may bias the evaluation results. The meta-frontier analysis [22,23] allows production technology heterogeneity to be taken into consideration. O'Donnell et al. [24] introduced this concept into DEA. According to the heterogeneity of production technology, all DMUs are divided into several groups accordingly. The production technologies of all the DMUs in the same group are identical. The meta-frontier is the envelope of all group frontiers. In the meta-frontier analysis, each DMU is assessed with respect to the corresponding group frontier and the meta-frontier. Then, a performance comparison across groups can be performed, and the technology gap between the group frontier and the meta-frontier can be calculated.

Heterogeneity and the meta-frontier analysis are popular topics in existing DEA studies. By considering regional technology heterogeneity, many studies divide groups to study the eco-efficiency of provinces in China [25,26]. As the study of meta-frontiers has deepened, scholars have realized that if there is still technology production heterogeneity in a group, the group should be divided into several subgroups. Most current studies considering dual heterogeneity focus on both regional and industrial heterogeneity. Feng et al. [26] proposed a three-hierarchy meta-frontier approach to study the energy efficiency of three industries (primary, secondary and tertiary) in three China's regions (Eastern, Central and Western). By applying the three-hierarchy meta-frontier DEA model and the panel data from 2001 to 2018, Tian and Feng [27] measured five key internal factors of China's green total-factor productivity.

Wang et al. [28] pointed out that the traditional meta-frontier analysis might generate the technical gap ratio (TGR) greater than 1 and thus violates the basic property of the meta-frontier analysis. To overcome this issue, they proposed a segmented projection approach based on the non-radial directional distance function. On this basis, Chen et al. [29] introduced a three-level meta-frontier SBM approach to evaluate the total factor energy efficiency in the Chinese manufacturing industry by considering dual heterogeneity and dividing the projection path into three segments. Compared with the study of Wang et al. [28], the method proposed by Chen et al. [29] considers three frontiers brought by dual heterogeneity and the inefficiency measured by slacks. See [30–35] for more studies addressing TGR > 1. Among them, only Yu and Rakshit [34] study uses network DEA [36]

to analyze the efficiency of internal divisions in depth, and other studies are constructed using "black-box" DEA models and thus cannot measure the performance of divisions inside the system.

Network DEA considers the data information of DMUs in each production division so that it can measure the performance of the whole system and all the divisions. In view of this advantage, many scholars have studied eco-efficiency by using network DEA [37]. Wang et al. [38] divided ecosystems into the production stage and the governance stage, and then, to analyze the technological heterogeneity in reducing pollution, they proposed a two-stage meta-frontier method by using network DEA [36]. More detailed meta-frontier studies considering stage division can be seen from Yu and Chen [39], although these studies cannot guarantee that the TGRs are not greater than 1.

Existing meta-frontier studies on eco-efficiency lack a comprehensive perspective that integrates considerations of regional and temporal heterogeneity, along with the internal network structure of ecosystems. This paper introduces a novel approach to studying eco-efficiency using meta-frontier analysis, incorporating stage division, dual heterogeneity and segmented projection. The key contributions of this research can be highlighted as follows:

**Integration of regional and temporal heterogeneities:** The paper makes a significant contribution by simultaneously incorporating regional and temporal technology heterogeneities. This approach enables the definition of group and subgroup frontiers, offering a more nuanced understanding of eco-efficiency.
**Stage-divided analysis:** By dividing eco-activities into distinct economic production and environmental governance stages, the research introduces a valuable perspective. This stage-oriented analysis facilitates the development of a two-stage three-level meta-frontier network SBM (NSBM) model, providing insights into eco-efficiency and stage efficiencies.
**Tackling TGR issue:** The proposed two-stage three-level meta-frontier model addresses a critical issue in meta-frontier methods, specifically handling instances where the TGRs of certain indexes exceed 1. This innovation contributes to the robustness and applicability of meta-frontier analysis.
**Comprehensive emissions inefficiency analysis:** The paper extends its focus to emissions inefficiency, treating pollutant emissions as intermediate variables connecting economic production and environmental governance stages. The introduced concept of total emissions inefficiency, encompassing management inefficiency, regional heterogeneity technology inefficiency and temporal heterogeneity technology inefficiency, provides a comprehensive framework for analyzing specific sources of emissions inefficiency.
**Regional and temporal TGR definition:** The research defines and discusses regional and temporal TGRs, offering a nuanced understanding of these ratios across 30 provinces in China. This contributes to a more granular evaluation of technological advancements and disparities in different regions over time.

The reminder of this paper is organized as follows: Section 2 introduces the two-stage three-level meta-frontier model ensuring TGRs not greater than 1 in the two-stage ecosystem. The emissions inefficiency index and its decomposition as well as the related TGRs are also introduced. In Section 3, we apply our method to evaluate the eco-efficiency of 30 provinces in China from 2016 to 2020 and analyze the influences from dual heterogeneity on eco-efficiency and emissions inefficiency. Section 4 offers conclusions and future work.

## 2. Methodology

### 2.1. Three-Level Meta-Frontier with the Two-Stage Structure

With the development of science and technology in economic production and environmental governance, there are gaps in the technology levels of different periods. In addition, Oh [40] and Wang et al. [25] indicated that geographic location is a main source of production technology heterogeneity. Therefore, we take into account both the temporal technology heterogeneity and the regional technology heterogeneity. We consider three kinds of frontiers, the regional (sub-group) frontiers, the temporal (group) frontiers and the

meta-frontier. Assume that there are $N_G$ periods and each period $t$ ($t \in \{1, \ldots, N_G\}$) is considered a group. The set of DMUs located in period $t$ is denoted $G^t$. Let $\mathbb{G} = \bigcup_{t=1}^{N_G} G^t$. Clearly, $G^{t_1} \cap G^{t_2} = \varnothing$ ($\forall t_1, t_2 \in \{1, \ldots, N_G\}$, $t_1 \neq t_2$). Considering the regional technology heterogeneity, we further divide each group $G^t$ into $N_{SG}^t$ subgroups. The set of DMUs located in region $v$ and period $t$ is denoted by $SG_v^t$ ($v \in \{1, \ldots, N_{SG}^t\}$). Of course, $\bigcup_{v=1}^{N_{SG}^t} SG_v^t = G_t, \forall t$ and $SG_{v_1}^t \cap SG_{v_2}^t = \varnothing$ ($\forall v_1, v_2 \in \{1, \ldots, N_{SG}^t\}$, $v_1 \neq v_2$).

By referring to current studies [38,41,42], we divide the activities of each DMU into two stages, namely the economic production (EP) stage and the environmental governance (EG) stage. In the EP stage, each DMU$_j$ ($j \in SG_v^t$, $G^t$ or $M$) consumes resources ($x_{ij}^1$, $i \in \{1, \ldots, m_1\}$), such as capital and energy, to obtain economic benefits ($y_{dj}^1, \forall d \in \{1, \ldots, r_1\}$) while being accompanied by pollutant emissions ($z_{hj}, \forall h \in \{1, \ldots, l\}$), such as $SO_2$, $CO_2$ and other emissions. In the EG stage, the environmental governance fund ($x_{qj}^2, \forall q \in \{1, \ldots, m_2\}$) is used to govern pollutant emissions generated from the EP stage. We think that there are two kinds of environmental governance results. One is effective governance ($y_{wj}^2$, $\forall w \in \{1, \ldots, r_2\}$) that improves the environment, and the other is governed ineffectively, which causes economic losses ($b_{1j}^2, r \in \{1, \ldots, g\}$). The detailed two-stage network structure is shown in Figure 1.

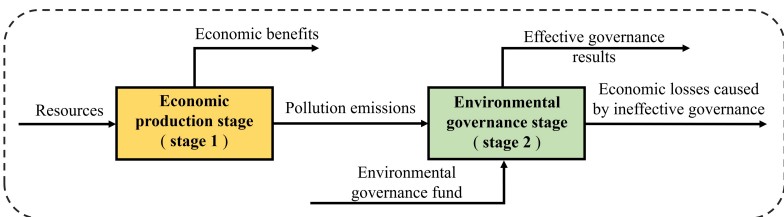

**Figure 1.** The network structure of the provincial two-stage eco-efficiency.

Figure 2 gives the three-segment projection path in the two-stage structure. The left-hand side of Figure 2 depicts the frontiers of the EP stage, where the vertical axis represents inputs $X^1$ or undesirable intermediate variable $Z$, and the horizontal axis represents outputs $Y^1$. Note that the intermediate variables in this paper are pollutant emissions. According to Liu et al. [43], they are the undesirable outputs of stage 1 and the undesirable inputs of stage 2. Therefore, whether in stage 1 or 2, decision-makers hope that the fewer intermediate variables, the better. The specific piecewise path that a DMU (DMU$\in SG_v^t, SG_v^t \subset G^t, G^t \subset \mathbb{G}$) projects to the meta-frontier of the EP stage (i.e., $PM$ in Figure 2) is as follows. First, the DMU starts from itself (point A) and projects to the regional frontier of the EP stage (i.e., $PS_v^t$ in Figure 2), and the projection point is determined as point B. Since the period and region of the DMUs in each subgroup are the same, its production efficiency depends on its own management. So, we can evaluate the *management production efficiency* (MPE) of point A based on the regional frontier $PS_v^t$. Then, we project point B to the temporal frontier (i.e., $PG^t$ in Figure 2) and denote the project point as point C. Since point B is already on its corresponding regional frontier, and inefficiencies brought by its management level have been eliminated, its production efficiency depends on the regional technology. Then, we can obtain the *regional production efficiency* (RPE) of point A based on the temporal frontier $PG^t$ and the projection point B. The line segment BC reflects the regional TGR between the regional frontier and the temporal frontier. Finally, we project point C to the meta-frontier $PM$ and denote the projection point as point D. Point C is already on its corresponding temporal frontier, and inefficiencies brought by its management level and regional technology have been eliminated. Since technology continues to change over time, we believe that in this projection path, the production efficiency of point C is only affected by temporal heterogeneity. Therefore, we can obtain the *temporal production efficiency* (TPE)

of point A based on the meta-frontier *PM* and the projection point C. Similarly, the line segment CD reflects the temporal TGR between the temporal frontier and the meta-frontier.

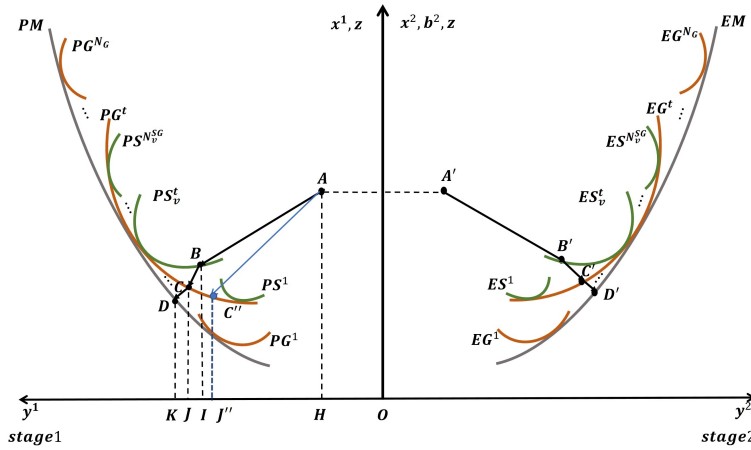

**Figure 2.** The three-level meta-frontier with the two-stage network structure.

The right-hand side of Figure 2 shows the frontiers of the EG stage, where the vertical axis represents inputs $X^2$, undesirable intermediate variable $Z$ or undesirable outputs $B^2$, and the horizontal axis represents desirable outputs $Y^2$. Point $A'$ is the point corresponding point A in the EG stage. The piecewise path that DMU projects to the meta-frontier of the EG stage (i.e., *EM* in Figure 2) is $A' \rightarrow B' \rightarrow C' \rightarrow D'$. The *management governance efficiency* (MGE), the *regional governance efficiency* (RGE) and the *temporal governance efficiency* (TGE) of the DMU, corresponding to projection paths $A'B'$, $B'C'$ and $C'D'$, can also be obtained in a similar way. For ease of description, we refer to the eco-efficiencies of the whole two-stage system corresponding to the three-segment projections, *management eco-efficiency* (ME), *regional eco-efficiency* (RE) and *temporal eco-efficiency* (TE).

This segmented projection allows the TGRs relative to variables to range between 0 and 1 in the two-stage ecosystem. Let us take the variable $y^1$ in Figure 2 as an example. According to Chen et al. [29], the efficiency of $y^1$ relative to the sub-group frontier, the group frontier and the meta-frontier can be defined as:

$$EM^{y^1} = \frac{\text{OH}}{\text{OI}},$$
$$ER^{y^1} = \frac{\text{OH}}{\text{OI} + \text{IJ}},$$
$$ET^{y^1} = \frac{\text{OH}}{\text{OI} + \text{IJ} + \text{JK}},$$

respectively, where $H$, $I$, $J$ and $K$ are the projection of points A, B, C and D onto the $y^1$ axis. For $y^1$, the TGR between the sub-group frontier and the group frontier as well as that between the group frontier and the meta-frontier can be expressed as

$$RTGR^{y^1} = \frac{ER^{y^1}}{EM^{y^1}} = \frac{\text{OI}}{\text{OI} + \text{IJ}},$$
$$TTGR^{y^1} = \frac{ET^{y^1}}{ER^{y^1}} = \frac{\text{OI} + \text{IJ}}{\text{OI} + \text{IJ} + \text{JK}}.$$

Clearly, the above two TGRs range between 0 and 1. The TGRs relative to other variables also have this property. Therefore, the basic property of the meta-frontier analysis can be guaranteed in the two-stage ecosystem.

However, if we adopt direct projection, this basic property might be avoided. We still take the variable $y^1$ in Figure 2 as an example. If the direct projection of point A on the

temporal frontier $PG^t$ is point C″ and the projection of point C″ onto the $y^1$ axis is point I″, then the efficiency of $y^1$ relative to the group frontier is $\frac{\text{OH}}{\text{OJ}''}$. As a result, the corresponding TGR between the sub-group frontier and the group frontier is expressed as $\frac{\text{OH/OJ}''}{\text{OH/OI}} = \frac{\text{OI}}{\text{OJ}''}$, which is greater than 1 since $\text{OI} > \text{OJ}''$.

With consideration of the above stage division and dual technology heterogeneity, we divide the path of projecting the target DMU to the meta-frontier into three segments for each stage. Then, we can easily explore which factors in which stage lead to the ecological inefficiency and emissions inefficiency of DMUs.

### 2.2. Three-Level Meta-Frontier NSBM Model

NSBM [44] is a non-radial NDEA model and is suitable for measuring efficiencies when inputs and outputs may change non-proportionally. Compared with the original NSBM proposed by Tone and Tsutsui [44], Kao [45]'s NSBM model does not need to specify the weight for each stage beforehand, and its system efficiency can be decomposed into the weighted average of stage efficiencies. Hence, we chose Kao [45]'s NSBM model in the design of our model. Using the segmented projection presented in Section 2.1, we built the following NSBM model to assess the overall efficiency of the target DMU$_o$ ($o \in SG_v^t, SG_v^t \subset G^t, G^t \subset \mathbb{G}, \forall v, t$):

$$E_{op}^* = \min \frac{1 - \frac{1}{m_1}\sum\limits_{i=1}^{m_1}\frac{s_{iop}^{1,x-}}{x_{iop}^1} + 1 - \frac{1}{m_2+l}\left(\sum\limits_{q=1}^{m_2}\frac{s_{qop}^{2,x-}}{x_{qop}^2} + \sum\limits_{h=1}^{l}\frac{s_{hop}^z}{z_{hop}}\right)}{1 + \frac{1}{r_1+l}\left(\sum\limits_{d=1}^{r_1}\frac{s_{dop}^{1,y+}}{y_{dop}^1} + \sum\limits_{h=1}^{l}\frac{s_{hop}^z}{z_{hop}}\right) + 1 + \frac{1}{r_2+g}\left(\sum\limits_{w=1}^{r_2}\frac{s_{wop}^{2,y+}}{y_{wop}^2} + \sum\limits_{r=1}^{g}\frac{s_{rop}^{2,b-}}{b_{rop}^2}\right)}$$

$$\text{s.t.} \quad x_{iop}^1 = \sum\limits_{j\in\Omega}\lambda_j^1 x_{ij}^1 + s_{iop}^{1,x-}, i = 1,\ldots,m_1,$$
$$y_{dop}^1 = \sum\limits_{j\in\Omega}\lambda_j^1 y_{dj}^1 - s_{dop}^{1,y+}, d = 1,\ldots,r_1,$$
$$z_{hop} = \sum\limits_{j\in\Omega}\lambda_j^1 z_{hj} + s_{hop}^z, h = 1,\ldots,l,$$
$$x_{qop}^2 = \sum\limits_{j\in\Omega}\lambda_j^2 x_{qj}^2 + s_{qop}^{2,x-}, q = 1,\ldots,m_2,$$
$$y_{wop}^2 = \sum\limits_{j\in\Omega}\lambda_j^2 y_{wj}^2 - s_{wop}^{2,y+}, w = 1,\ldots,r_2, \qquad (1)$$
$$b_{rop}^2 = \sum\limits_{j\in\Omega}\lambda_j^2 b_{rj}^2 + s_{rop}^{2,b-}, r = 1,\ldots,g,$$
$$z_{hop} = \sum\limits_{j\in\Omega}\lambda_j^2 z_{hj} + s_{hop}^z, h = 1,\ldots,l,$$
$$\sum\limits_{j\in\Omega}\lambda_j^1 z_{hj} = \sum\limits_{j\in\Omega}\lambda_j^2 z_{hj}, h = 1,\ldots,l,$$
$$\lambda_j^1, \lambda_j^2 \geq 0, j \in \Omega,$$
$$s_{iop}^{1,x-}, s_{dop}^{1,y+}, s_{hop}^z, s_{qop}^{2,x-}, s_{wop}^{2,y+}, s_{rop}^{2,b-} \geq 0, \forall i, d, h, q, w, r,$$

where $p$ and $\Omega$ have three sets of values, corresponding to three projection paths. If $p = R$, then $\Omega = SG_v^t$, which means that the optimal value of model (1), $E_{oR}^*$, is equal to the ME of DMU$_o$, i.e., the two-stage system eco-efficiency of DMU$_o$ with respect to the regional frontier. If $p = T$, then $\Omega = G^t$, which means that model (1) is used to calculate the RE of DMU$_o$, i.e., the two-stage system eco-efficiency of the projection of DMU$_o$ on the regional frontier with respect to the temporal frontier. If $p = M$, then $\Omega = \mathbb{G}$, which means that model (1) is used to calculate the TE of DMU$_o$, i.e., the two-stage system eco-efficiency of the projection of DMU$_o$ on the temporal frontier with respect to the meta-frontier. Moreover, in model (1),

$$x_{ioR}^1 = x_{io}^1, \; y_{doR}^1 = y_{do}^1, \; z_{hoR} = z_{ho}, \; x_{ioR}^2 = x_{io}^2, \; y_{woR}^2 = y_{wo}^2, \; b_{roR}^2 = b_{ro}^2,$$
$$x_{ioT}^1 = x_{io}^1 - s_{iR}^{1,x-*}, \; y_{doT}^1 = y_{do}^1 + s_{dR}^{1,y+*}, \; z_{hoT} = z_{ho} - s_{hR}^{z*},$$
$$x_{qoT}^2 = x_{qo}^2 - s_{qR}^{2,x-*}, \; y_{woT}^2 = y_{wo}^2 + s_{wR}^{2,y+*}, \; b_{roT}^2 = b_{ro}^2 - s_{rR}^{2,b-*},$$

$$x^1_{ioM} = x^1_{io} - s^{1,x-*}_{iR} - s^{1,x-*}_{iT}, \ y^1_{doM} = y^1_{do} + s^{1,y+*}_{dR} + s^{1,y+*}_{dT},$$

$$z_{hoM} = z_{ho} - s^{z*}_{hR} - s^{z*}_{hT}, \ x^2_{qoM} = x^2_{qo} - s^{2,x-*}_{qR} - s^{2,x-*}_{qT},$$

$$y^2_{woM} = y^2_{wo} + s^{2,y+*}_{wR} + s^{2,y+*}_{wT}, \ b^2_{roM} = b^2_{ro} - s^{2,b-*}_{rR} - s^{2,b-*}_{rT}, \ \forall i, d, h, q, w, r,$$

where variables with an asterisk in the upper right corner are the optimal solution of model (1). In our model, we adopt the free links constraints $\sum_{j \in \Omega} \lambda^1_j z_{hj} = \sum_{j \in \Omega} \lambda^2_j z_{hj}, \forall h$ [44], so that TGRs and efficiency indexes with respect to the intermediate variables are same in two stages. So we just use $s^z_{hop}$ ($\forall h$) to express them in model (1).

Once we obtain an optimal solution of model (1), the efficiency scores for the two individual stages can be calculated as

$$\theta^1_{op} = \frac{1 - \frac{1}{m_1} \sum\limits_{i=1}^{m_1} \frac{s^{1,x-*}_{iop}}{x^1_{iop}}}{1 + \frac{1}{r_1 + l} \left( \sum\limits_{d=1}^{r_1} \frac{s^{1,y+*}_{dop}}{y^1_{dop}} + \sum\limits_{h=1}^{l} \frac{s^{z*}_{hop}}{z_{hop}} \right)},$$

$$\theta^2_{op} = \frac{1 - \frac{1}{m_2 + l} \left( \sum\limits_{q=1}^{m_2} \frac{s^{2,x-*}_{qop}}{x^2_{qop}} + \sum\limits_{h=1}^{l} \frac{s^{z*}_{hop}}{z_{hop}} \right)}{1 + \frac{1}{r_2 + g} \left( \sum\limits_{w=1}^{r_2} \frac{s^{2,y+*}_{wop}}{y^2_{wop}} + \sum\limits_{r=1}^{g} \frac{s^{2,b-*}_{rop}}{b^2_{rop}} \right)}, \ \forall p = R, T, M.$$

We can also obtain a set of weights as

$$\omega^1_{op} = \frac{1 + \frac{1}{r_1 + l} \left( \sum\limits_{d=1}^{r_1} \frac{s^{1,y+*}_{dop}}{y^1_{dop}} + \sum\limits_{h=1}^{l} \frac{s^{z*}_{hop}}{z_{hop}} \right)}{1 + \frac{1}{r_1 + l} \left( \sum\limits_{d=1}^{r_1} \frac{s^{1,y+*}_{dop}}{y^1_{dop}} + \sum\limits_{h=1}^{l} \frac{s^{z*}_{hop}}{z_{hop}} \right) + 1 + \frac{1}{r_2 + g} \left( \sum\limits_{w=1}^{r_2} \frac{s^{2,y+*}_{wop}}{y^2_{wop}} + \sum\limits_{r=1}^{g} \frac{s^{2,b-*}_{rop}}{b^2_{rop}} \right)},$$

$$\omega^2_{op} = \frac{1 + \frac{1}{r_2 + g} \left( \sum\limits_{w=1}^{r_2} \frac{s^{2,y+*}_{wop}}{y^2_{wop}} + \sum\limits_{r=1}^{g} \frac{s^{2,b-*}_{rop}}{b^2_{rop}} \right)}{1 + \frac{1}{r_1 + l} \left( \sum\limits_{d=1}^{r_1} \frac{s^{2,y+*}_{dp}}{y^1_{dop}} + \sum\limits_{h=1}^{l} \frac{s^{z*}_{hop}}{z_{hop}} \right) + 1 + \frac{1}{r_2 + g} \left( \sum\limits_{w=1}^{r_2} \frac{s^{2,y+*}_{wop}}{y^2_{wop}} + \sum\limits_{r=1}^{g} \frac{s^{2,b-*}_{rop}}{b^2_{rop}} \right)}. \tag{2}$$

Clearly, we have

$$E^*_{op} = \theta^1_{op} * \omega^1_{op} + \theta^2_{op} * \omega^2_{op}. \tag{3}$$

However, model (1) might have multiple optimal solutions so that the values of $\theta^1_{op}$ and $\theta^2_{op}$ might not be unique. Therefore, we follow the procedure adopted by Kao and Hwang [46] and Chen et al. [47] to determine the highest efficiency score of stage 1 or 2 while maintaining the efficiency score of the whole two-stage system. Considering that, in the real world, economic production is a prerequisite for environmental governance, we first determine the efficiency of stage 1 and then calculate that of stage 2. Then, we have

$$\theta^{1*}_{op} = \min \frac{1 - \frac{1}{m_1} \sum\limits_{i=1}^{m_1} \frac{s^{1,x-}_{iop}}{x^1_{iop}}}{1 + \frac{1}{r_1 + l} \left( \sum\limits_{d=1}^{r_1} \frac{s^{1,y+}_{dop}}{y^1_{dop}} + \sum\limits_{h=1}^{l} \frac{s^z_{hop}}{z_{hop}} \right)}$$

**s.t.** $\quad x^1_{iop} = \sum\limits_{j \in \Omega} \lambda^1_j x^1_{ij} + s^{1,x-}_{iop}, i = 1, \dots, m_1,$

$$y^1_{dop} = \sum\limits_{j \in \Omega} \lambda^1_j y^1_{dj} - s^{1,y+}_{dop}, d = 1, \dots, r_1,$$

$$z_{hop} = \sum\limits_{j \in \Omega} \lambda^1_j z_{hj} + s^z_{hop}, h = 1, \dots, l,$$

$$x_{qop}^2 = \sum_{j \in \Omega} \lambda_j^2 x_{qj}^2 + s_{qop}^{2,x-}, q = 1, \ldots, m_2,$$

$$y_{dop}^2 = \sum_{j \in \Omega} \lambda_j^2 y_{wj}^2 - s_{wop}^{2,y+}, w = 1, \ldots, r_2,$$

$$b_{ro}^{2p} = \sum_{j \in \Omega} \lambda_j^2 b_{rj}^2 + s_{rop}^{2,b-}, r = 1, \ldots, g, \tag{4}$$

$$z_{hop} = \sum_{j \in \Omega} \lambda_j^2 z_{qj} + s_{hop}^z, h = 1, \ldots, l,$$

$$\sum_{j \in \Omega} \lambda_j^1 z_{hj} = \sum_{j \in \Omega} \lambda_j^2 z_{hj}, h = 1, \ldots, l,$$

$$2 - \frac{1}{m_1} \sum_{i=1}^{m_1} \frac{s_{iop}^{1,x-}}{x_{iop}^1} - \frac{1}{m_2+l} \left( \sum_{q=1}^{m_2} \frac{s_{qop}^{2,x-}}{x_{qop}^2} + \sum_{h=1}^{l} \frac{s_{hop}^z}{z_{hop}} \right) = E_{op}^*$$

$$\left[ 2 + \frac{1}{r_1+l} \left( \sum_{d=1}^{r_1} \frac{s_{dop}^{1,y+}}{y_{dop}^1} + \sum_{h=1}^{l} \frac{s_{hop}^z}{z_{hop}} \right) + \frac{1}{r_2+g} \left( \sum_{w=1}^{r_2} \frac{s_{wop}^{2,y+}}{y_{wop}^2} + \sum_{r=1}^{g} \frac{s_{rop}^{2,b-}}{b_{rop}^2} \right) \right],$$

$$\lambda_j^1, \lambda_j^2 \geq 0, j \in \Omega,$$

$$s_{iop}^{1,x-}, s_{dop}^{1,y+}, s_{hop}^z, s_{qop}^{2,x-}, s_{wop}^{2,y+}, s_{rop}^{2,b-} \geq 0, \forall j, d, h, q, w, r.$$

We still let $s_{iop}^{1,x-*}, s_{dop}^{1,y+*}, s_{hop}^{z*}, s_{qop}^{2,x-*}, s_{wop}^{2,y+*}, s_{rop}^{2,b-*}$ ($\forall j, i, d, h, q, w, r$) be the optimal solution variables of model (4). Then, we can obtain $\omega_{op}^1$ and $\omega_{op}^2$ by (2). The efficiency score of stage 2 for DMU$_o$, i.e., $\theta_{op}^{2*}$, is calculated as

$$\theta_{op}^{2*} = \frac{E_{op}^* - \theta_{op}^{1*} * \omega_{op}^1}{\omega_{op}^2}, \tag{5}$$

where $p$ is identical to that in model (1). By solving models (1) and (4) for three different sets in the order $p = R, T, M$, we can obtain ME, MPE, RE, RPE, TE and TPE of DMU$_o$ as well as the corresponding optimal solutions. Thus, by calculating (2) and (5), we can obtain MGE, RGE and TGE.

*2.3. Emissions Inefficiency and TGRs*

Compared with other evaluation problems, one of the most significant characteristics of eco-efficiency evaluation is that it considers the adverse impact on the environment. In the two-stage ecosystem shown in Figure 1, this adverse impact is represented by pollution emissions. Therefore, in order to better analyze the two-stage ecosystem, we here study the inefficiency and TGRs of pollution emissions (i.e., intermediate variables).

In order to further explore the impact of management level on pollutant emissions, we define the *management emissions inefficiency* (MI) as the average ratio of the slacks of intermediate variables in model (4) corresponding to the first projection path to the sample, i.e.,

$$MI = \frac{1}{h} \sum_{h=1}^{l} \frac{s_{hoR}^{z*}}{z_{ho}}. \tag{6}$$

To further explore the impact of regional development level on emissions inefficiency, we define the *regional heterogeneity technology emissions inefficiency* (RHI) as the average ratio of the slacks of intermediate variables in model (4) corresponding to the second projection path to the sample, i.e.,

$$RHI = \frac{1}{h} \sum_{h=1}^{l} \frac{s_{hoT}^{z*}}{z_{ho}}. \tag{7}$$

By referring to Chen et al. [29], we define the *regional TGR with respect to emissions* (RTGR) as the average ratio of the project values of the intermediate variables on the temporal frontier to those on the regional frontier, i.e.,

$$RTGR = \frac{1}{h} \sum_{h=1}^{l} \frac{z_{ho} - s_{hoT}^{z*} - s_{hoR}^{z*}}{z_{ho} - s_{hoR}^{z*}} = \frac{1}{h} \sum_{h=1}^{l} \frac{z_{hoM}}{z_{hoT}}. \tag{8}$$

RTGR reflects the gap between the emissions technology level of a specific region and that of the temporal frontier. The higher the RTGR, the closer the two emissions technology levels.

To further explore the impact of different temporal technology levels on emissions inefficiency, we define the *temporal heterogeneity technology emissions inefficiency* (THI) as the average ratio of the slacks of the intermediate variables in model (4) corresponding to the third projection path to the sample, i.e.,

$$THI = \frac{1}{h} \sum_{h=1}^{l} \frac{s_{hoM}^{z*}}{z_{ho}}. \tag{9}$$

Similarly, we define the *temporal TGR with respect to emissions* (TTGR) as the average ratio of the project values of intermediate variables on the meta-frontier to those on the temporal frontier. Namely,

$$TTGR = \frac{1}{h} \sum_{h=1}^{l} \frac{z_{ho} - s_{hoR}^{z*} - s_{hoT}^{z*} - s_{hoM}^{z*}}{z_{ho} - s_{hoR}^{z*} - s_{hoT}^{z*}} = \frac{1}{h} \sum_{h=1}^{l} \frac{z_{hoM} - s_{hoM}^{z*}}{z_{hoM}}. \tag{10}$$

TTGR reflects the gap between the emissions technology level of the temporal frontier and that of the meta-frontier. Similar, the higher the TTGR, the closer the two emissions technology levels.

According to Chen et al. [29], we define the *total emissions inefficiency* (TEI) as

$$TEI = MI + RHI + THI = \frac{1}{h} \sum_{h=1}^{l} \frac{s_{hoR}^{z*} + s_{hoT}^{z*} + s_{hoM}^{z*}}{z_{ho}}. \tag{11}$$

## 3. Empirical Analysis

Here, all the models were coded with PyCharm 2019.3.1 (Community Edition) software, combined with Gurobi and the basic library of Python.

### 3.1. Data and Indexes

For simplicity, we will refer to provinces, autonomous regions and municipalities as provinces. We treat provinces as DMUs. Due to the lack of energy data in Tibet and the differences in statistical methods of Hong Kong and Macao, we only analyzed 30 provinces in mainland China. Considering that the 13th Five-Year Plan has just been implemented, we are concerned about the eco-efficiency during its implementation. So, the sampled period is from 2016 to 2020. A calendar year is treated as one group. In each sample year, we divided 30 provinces into the eastern, central and western regions with reference to the three major regions (subgroup) divided by the National Bureau of Statistics of China. The specific division is shown in Table 2.

**Table 2.** Three major regions in China.

| Region | Province |
| --- | --- |
| Eastern | Beijing, Tianjin, Shanghai, Shandong, Hebei, Jiangsu, Zhejiang, Fujian, Guangdong, Hainan, Liaoning |
| Central | Jilin, Heilongjiang, Henan, Shanxi, Anhui, Hubei, Hunan, Jiangxi |
| Western | Chongqing, Sichuan, Guizhou, Yunnan, Guangxi, Shaanxi, Gansu, Qinghai, Inner Mongolia, Ningxia, Xinjiang |

Most relevant studies select capital, energy and labor or population as resource inputs. See Table 1 for details. By referring to them, the fixed asset investment was chosen as the capital input and the resident population at year-end was selected as the population input in this study. Since the main energy consumption in China is still coal, the coal consumption was selected as the energy input. We chose the gross regional product (GDP) as the economic benefit. For intermediate variables, although there are a large number of pollutant emissions to choose, too many indicators will reduce the efficiency discrimination power of DEA. Considering that most relevant studies [12–16,18–20] choose $SO_2$ and/or $CO_2$ emissions, we regard $SO_2$ and $CO_2$ emissions as two intermediate variables. We chose the pollution control investment as an exogenous input of the EG stage. Good air days can reflect an effective environmental governance effect. Since the capital city and central cities of a province are usually the focus of environmental governance attention of the government, we used the average proportion of the number of days with high quality air (e.g., the days with an air quality index (AQI) of no more than 100) in these cities to the total number of days observed in the whole year as an effective governance index. In addition, the direct economic loss caused by meteorological disasters is regarded as an undesirable output to represent the results of ineffective governance.

Table 3 introduces the data source. The index data are directly or indirectly obtained from the resources shown in Table 3. Some remarks on the data are presented here.

**Table 3.** Data resources.

| Indicator | Index | Resource |
| --- | --- | --- |
| Resident population at year-end | $x_1^1$ | National Bureau of Statistics of China |
| Coal consumption | $x_2^1$ | National Bureau of Statistics of China |
| Fixed asset investment | $x_3^1$ | China Economic and Social Big Data Research Platform |
| GDP | $y_1^1$ | National Bureau of Statistics of China |
| $SO_2$ emissions | $z_1$ | National Bureau of Statistics of China |
| $CO_2$ emissions | $z_2$ | China Carbon Accounting Database (https://www.ceads.net.cn, accessed on 10 February 2022) |
| Pollution control investment | $x_1^2$ | National Bureau of Statistics of China |
| Proportion of high-quality air days | $y_1^2$ | Zhenqi network (https://www.zq12369.com, accessed on 10 February 2022) |
| Direct economic loss | $b_1^2$ | China Meteorological Disaster Yearbook |

(1) The data of fixed asset investment from 2016 to 2017 are directly given by the China Economic and Social Big Data Research Platform, while those from 2018 to 2020 are calculated using the "fixed asset investment growth over the previous year" index. This is because the index of fixed asset investment has not been published since 2018, only fixed asset investment growth over the previous year.

(2) $y_1^2$ is calculated using AQI indexes whose data are derived from the Zhenqi network.

(3) We estimate carbon emissions of provinces using the method proposed by the Intergovernmental Panel on Climate Change (IPCC), which is an internationally acknowledged method for accounting for carbon emissions and has been recognized by many scholars.

According to the actual situation and data characteristics, the projection of $y_1^2$ on the each frontier should not exceed 1. So, we need to add additional constraints in models (1) and (4). For $p = R$, $\Omega = SG_v^t$, we add $y_{wo}^2 + s_{woR}^{2,y+} \leq 1, w = 1, \ldots, r_2$; for $p = T$, $\Omega = G^t$, we add $y_{wo}^2 + s_{woR}^{2,y+*} + s_{woT}^{2,y+} \leq 1, w = 1, \ldots, r_2$; for $p = M$, $\Omega = \mathbb{G}$, we add $y_{wo}^2 + s_{woR}^{2,y+*} + s_{woT}^{2,y+*} + s_{woM}^{2,y+} \leq 1, w = 1, \ldots, r_2$.

Table 4 presents the descriptive statistics of all variables from 2016 to 2020. We can see that the average energy consumption in China maintains an increasing trend from 2016 to 2020. The average annual growth rate of energy consumption increases by 2.42% and 3.48% in 2017 and 2018, respectively, and then the growth rate slows down. Actually, except for $x_1^1$ showing a downward trend, the averages of inputs and outputs in the EP stage increase year-by-year. The average SO$_2$ emissions show a clear downward trend, and the standard deviation among provinces is shrinking, while average CO$_2$ emissions grow slowly overall. Except $x_1^2$, whose average generally shows a downward trend, the averages of input and output variables in the EG stage show significant fluctuating trends. All the above indicates that it is meaningful to discuss the gap among China's regional and temporal eco-efficiencies.

**Table 4.** Descriptive statistics of variables.

| | $x_1^1$ ($10^6$ Persons) | $x_2^1$ ($10^6$ Tons) | $x_3^1$ ($10^8$ RMB) | $y_1^1$ ($10^8$ RMB) | $z_1$ ($10^4$ Tons) | $z_2$ ($10^7$ RMB) | $x_1^2$ ($10^8$ RMB) | $y_1^2$ (%) | $b_1^2$ ($10^8$ RMB) |
|---|---|---|---|---|---|---|---|---|---|
| **2016** | | | | | | | | | |
| AVG | 26.41 | 141.65 | 19.90 | 24.99 | 28.49 | 40.88 | 27.30 | 0.72 | 166.67 |
| STD | 16.38 | 105.47 | 12.74 | 19.55 | 18.50 | 30.65 | 26.09 | 0.16 | 194.79 |
| Max | 67.03 | 409.39 | 53.32 | 82.16 | 72.98 | 142.06 | 126.41 | 0.99 | 837.70 |
| Min | 3.24 | 8.48 | 3.53 | 2.26 | 1.34 | 5.50 | 1.61 | 0.43 | 0.20 |
| **2017** | | | | | | | | | |
| AVG | 26.40 | 145.07 | 20.61 | 27.69 | 20.35 | 41.84 | 22.72 | 0.71 | 100.03 |
| STD | 16.46 | 109.75 | 14.49 | 21.65 | 12.71 | 31.31 | 22.78 | 0.15 | 126.11 |
| Max | 68.58 | 429.42 | 55.20 | 91.65 | 43.31 | 145.53 | 113.10 | 0.98 | 588.00 |
| Min | 3.27 | 4.90 | 1.98 | 2.47 | 0.65 | 5.16 | 1.53 | 0.41 | 0.00 |
| **2018** | | | | | | | | | |
| AVG | 26.32 | 150.12 | 21.84 | 30.42 | 17.19 | 43.01 | 20.71 | 0.81 | 87.90 |
| STD | 16.49 | 123.11 | 15.83 | 23.57 | 10.96 | 32.16 | 23.44 | 0.14 | 88.34 |
| Max | 69.60 | 489.40 | 57.47 | 99.95 | 36.33 | 144.03 | 98.75 | 1.00 | 340.50 |
| Min | 3.29 | 2.76 | 2.18 | 2.75 | 0.27 | 5.00 | 0.36 | 0.49 | 0.90 |
| **2019** | | | | | | | | | |
| AVG | 26.15 | 154.09 | 23.07 | 32.69 | 15.23 | 44.52 | 20.51 | 0.77 | 108.97 |
| STD | 16.47 | 129.67 | 16.66 | 25.24 | 9.32 | 33.45 | 20.23 | 0.14 | 136.10 |
| Max | 69.95 | 513.32 | 58.77 | 107.99 | 35.24 | 147.13 | 95.43 | 0.99 | 552.60 |
| Min | 3.30 | 1.83 | 2.24 | 2.94 | 0.19 | 4.91 | 0.63 | 0.49 | 0.00 |
| **2020** | | | | | | | | | |
| AVG | 24.96 | 154.67 | 23.76 | 33.60 | 10.59 | 44.90 | 15.14 | 0.82 | 123.25 |
| STD | 16.32 | 132.13 | 17.06 | 26.04 | 6.41 | 34.13 | 14.57 | 0.12 | 135.97 |
| Max | 70.39 | 537.39 | 58.94 | 111.15 | 27.39 | 145.35 | 53.13 | 1.00 | 602.60 |
| Min | 2.79 | 1.35 | 2.33 | 3.01 | 0.18 | 4.58 | 0.05 | 0.56 | 0.90 |

## 3.2. Eco-Efficiency Affected by Dual Heterogeneities

### 3.2.1. Management Eco-Efficiency

In the first level projection path, we obtain the results of ME, MPE and MGE. Figure 3 shows their trends in each region and China from 2016 to 2020. In the eastern region, the MPE is better than the MGE in 2016 and 2020, and both of them are almost equal from 2017 to 2019. In the central region, except 2017, the MPE is better than the MGE in the remaining years. In the western region and the whole of China, the MPE is better than the MGE in the five years. Obviously, Figure 3 illustrates that the ME in the eastern region is the worst. In 2017, the ME in the central region shows a drastic rise. In 2020, ME in each

region as well as the whole of China show a slight decline and show relatively flat changes from 2017 to 2019.

In order to better observe the differences of ME values in different regions and periods, we performed the Mann–Whitney U test [48] on ME in the regional and temporal dimensions. The Mann–Whitney U test is one of the most widely used statistical tests in behavioral research. It assumes that two samples are derived two populations that are identical except for the population mean and is intended to test whether the means of the two populations differ significantly. We utilized the *Scipy* package in Python for calculations and chose the null hypothesis $H_0: \mu_1 \geq \mu_2$ and the alternative hypothesis $H_1: \mu_1 < \mu_2$. The resulting *p* values in the regional dimension are provided in Table 5. The *p* values in Table 5 reveal that the ME in the central region is the highest, followed by that in the western region, and the worst is that in the eastern region. In the temporal dimension, only the ME in 2019 is better than that in 2020 with the relevant $p = 9.77\%$, which passes the 10% significance level test. There is no significant difference in ME of China in other years since all the relevant *p* values do not pass the 10% significance level test. For simplicity, we no longer provide these *p* values in the temporal dimension.

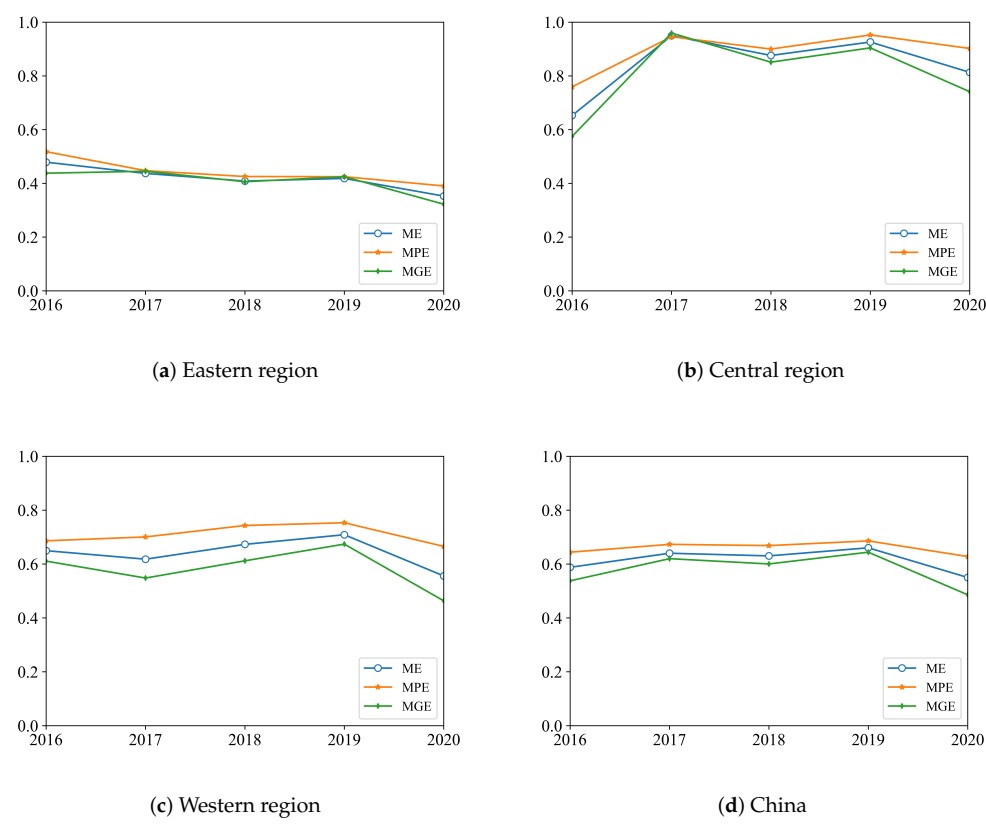

**Figure 3.** Trends of ME, MPE and MGE from 2016 to 2020.

**Table 5.** Mann–Whitney U test results of ME in the regional dimension.

| Region | Eastern | Central | Western |
|---|---|---|---|
| Eastern | - | **0** | **0** |
| Central | 1 | - | 0.9974 |
| Western | 1 | **0.0027** | - |

Figure 4 shows the annual average ME (AME) of each province. From Figure 4, the management level of the provinces in the eastern region is polarized. Beijing, Tianjin, Shanghai and Hainan have far higher AME than the remaining seven eastern provinces, among which Beijing and Shanghai are the provinces with the best AME in the eastern

region. Seven ME-inefficient provinces in the eastern region (i.e., Hebei, Liaoning, Jiangsu, Zhejiang, Fujian, Shandong, Guangdong) have very poor ME values. They should try their best to improve their management level in both economic production and environmental governance. On the whole, the central region has the highest AME and the smallest difference in AME among provinces. Jiangxi is the province with the best AME in the central region, which is always 1.0 during the sample period. Anhui and Gansu, as the provinces with the worst AME in the central and western regions, are still superior to many provinces in the eastern region.

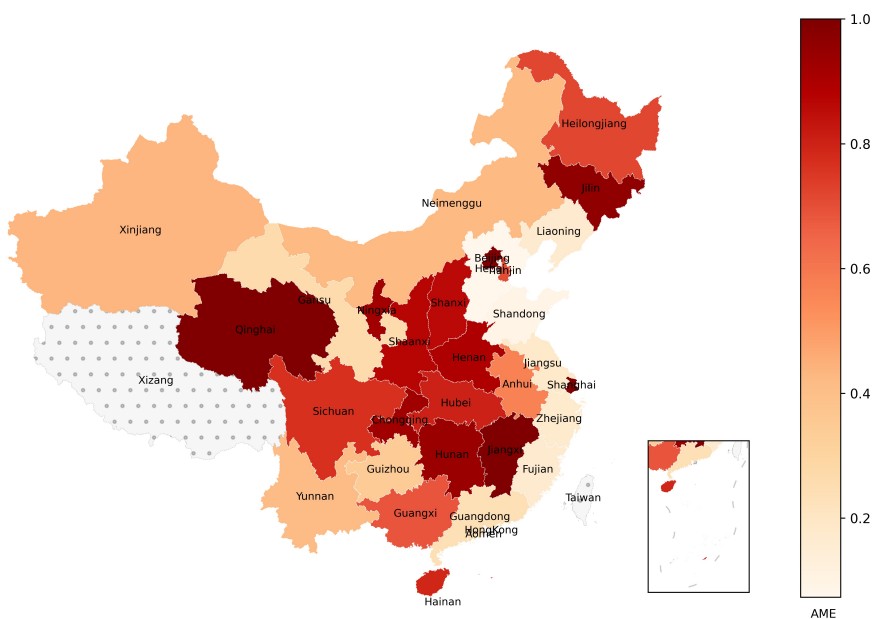

**Figure 4.** The AME of each province in China.

3.2.2. Regional Eco-Efficiency

In the second-level projection path, the results of RE, RPE and RGE can be obtained. Figure 5 shows the trends of RE, RPE and RGE in each region as well as the entirety of China from 2016 to 2020. It can be seen that in the central and western regions as well as in the whole of China, RPE is slightly better than RGE from 2018 to 2020. Specially, in the central region, RPE is better than RGE from 2016 to 2020. The trends of these three regional eco-efficiency indexes in the central and western regions as well as the whole of China are similar. From 2016 to 2018, they continue to decline, and from 2018 to 2020, they show flat changes. From Figure 5, we see the RE in the eastern region is the best, followed by that in the western region. The RE in the central region is the worst, which is less than 0.2 each year. Although most eastern provinces have very low ME, they are affected by the positive effects of excellent regional technology. Furthermore, most central provinces have good ME, but they are affected by the negative effect of poor regional technology.

We performed the Mann–Whitney U test on RE in both regional and temporal dimensions. In the regional dimension, the detailed $p$ values are shown in Table 6. This results show that the RE of the eastern region is the best, that of the western region is the second and that of the central region is the worst. In the temporal dimension, with the relevant $p = 9.53\%$ and $9.90\%$ passing the 10% significance level test, the RE in 2018 and 2019 is significantly lower than that in 2016. Other relevant $p$ values in the temporal dimension do not pass the 10% significance level. For simplicity, we do not provide them here. These results are consistent with those shown in Figure 5.

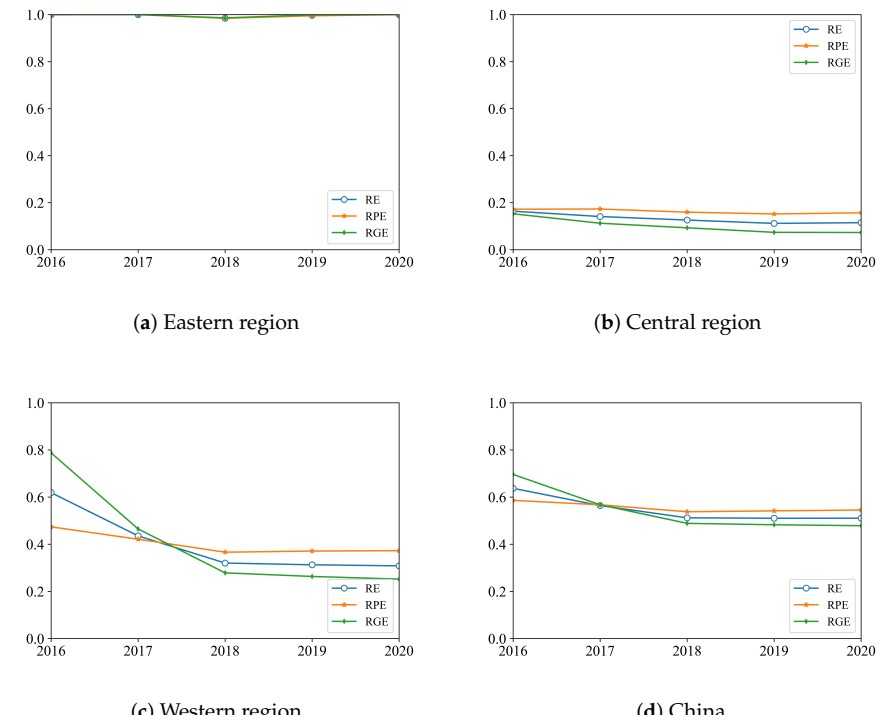

**Figure 5.** Trends of RE, RPE and RGE from 2016 to 2020.

**Table 6.** Mann–Whitney U test results of RE in the regional dimension.

| Region | Eastern | Central | Western |
|---|---|---|---|
| Eastern | - | 1 | 1 |
| Central | **0** | - | **0** |
| Western | **0** | 1 | - |

The RE of the eastern region is equal to or close to 1 every year. Only in 2016, 2018 and 2019 do several provinces not reach efficiency. Therefore, we observed the RE slack variables of these provinces (see Table 7) and analyzed the reasons why these provinces are not efficient. For Hebei and Fujian in 2016, the positive slack variables are concentrated on the inputs of the first stage. So, their regional eco-inefficiency mainly derives from the excessive inputs in the economic production stage. For Tianjin in 2018, except $x_1^1$, $y_1^2$ and $b_1^2$, there are excesses in the remaining inputs and intermediate products and insufficiency in the remaining outputs. For Jiangsu, Guangdong and Zhejiang in 2019, positive slacks are mainly located in the inputs of both stages and $z_2$ (i.e., the SO$_2$ emissions). Although there are positive slacks in the above provinces, their RE is still excellent. These provinces can focus on the above weaknesses to improve regional eco-efficiency. All eastern provinces, including the six provinces mentioned above, are regionally eco-efficient in 2017 and 2022.

**Table 7.** RE slacks of inefficient eastern provinces.

| | Year | $x_1^1$ | $x_2^1$ | $x_3^1$ | $y_1^1$ | $z_1$ | $z_2$ | $x_1^2$ | $y_1^2$ | $b_1^2$ |
|---|---|---|---|---|---|---|---|---|---|---|
| Hebei | 2016 | 0.024 | 0.027 | 0.037 | 0 | 0 | 0 | 0 | 0 | 0 |
| Fujian | 2016 | 0.019 | 0.022 | 0.029 | 0 | 0 | 0 | 0 | 0 | 0 |
| Tianjin | 2018 | 0 | 1.848 | 0.690 | 1.663 | 0.452 | 0.405 | 0.780 | 0 | 0 |
| Jiangsu | 2019 | 0.016 | 0.325 | 0.085 | 0 | 0.001 | 0.082 | 0.127 | 0 | 0 |
| Zhejiang | 2019 | 0.003 | 0.067 | 0.018 | 0 | $10^{-4}$ | 0.017 | 0.026 | 0 | 0 |
| Guangdong | 2019 | 0.019 | 0.388 | 0.102 | 0 | 0 | 0.098 | 0.142 | 0 | 0.010 |

### 3.2.3. Temporal Eco-Efficiency

In the third-level projection path, the results of TE, TPE and TGE are obtained. Figure 6 shows the treads of TE, TPE and TGE in the regions as well as the whole of China from 2016 to 2020. In the regions as well as the whole of China, TPE outperforms TGE in most years. This means that in most years, the EP performance depending on temporal technology is better than the EG performance depending on it in China. We find from Figure 6 that for the temporal eco-efficiency indexes, there is no significant difference among regions in China. The results of the Mann–Whitney U test on TE verify this. All the relevant *p* values in the regional dimension are greater than 0.3. This characteristic is different from that of the management eco-efficiency indexes and the regional eco-efficiency indexes, where some regions are significantly superior to others. The trends of the three temporal eco-efficiency indexes in the regions and the whole of China are similar. TPE shows a consistent upward trend in all regions and the country. TE and TGE show small declines in 2018 and both temporal eco-efficiency indexes rise steadily in the remaining years. The Mann–Whitney U test results in Table 8 also verify this. In Table 8, almost all of the upper triangle elements in the Mann–Whitney U test have passed the 5% significance test, which means that the TE of China increases in general. In 2020, all the three temporal efficiency indexes of each region and the entire country increase to scores close to 1.

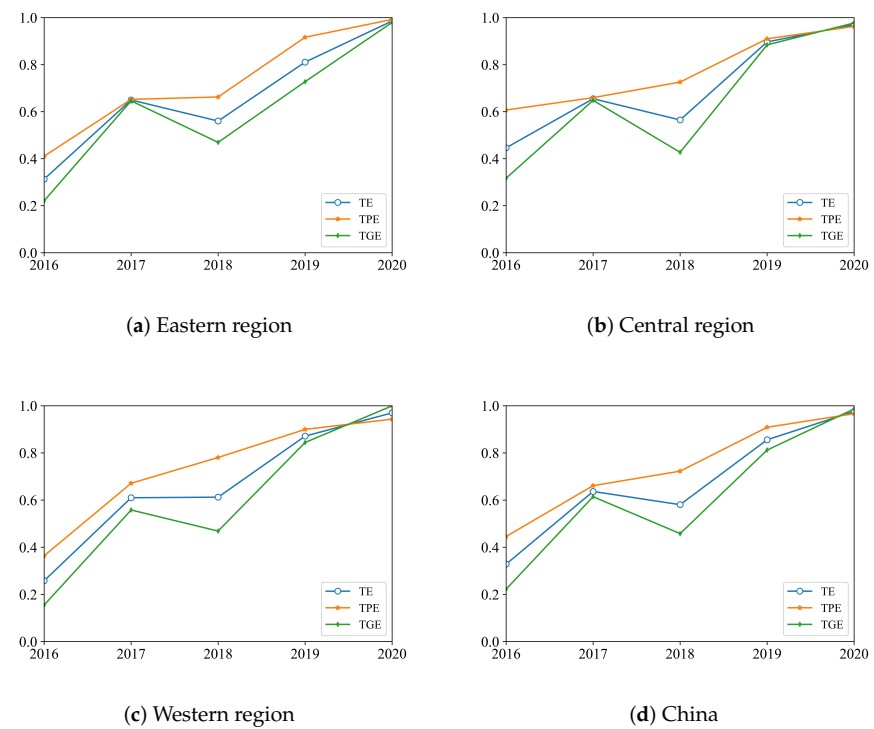

**Figure 6.** Trends of TE, TPE and TGE from 2016 to 2020.

**Table 8.** Mann–Whitney U test results of TE in the temporal dimension.

| Year | 2016 | 2017 | 2018 | 2019 | 2020 |
|------|------|------|------|------|------|
| 2016 | - | **0** | **0** | **0** | **0** |
| 2017 | 1 | - | 0.6247 | **0** | **0** |
| 2018 | 1 | 0.3809 | - | **0** | **0** |
| 2019 | 1 | 1 | 1 | - | 0.1199 |
| 2020 | 1 | 1 | 1 | 0.883 | - |

### 3.3. Emissions Analysis

3.3.1. Emissions Inefficiency and Its Decomposition

As an essential component of the ecosystem, the emissions of $SO_2$ and $CO_2$ serve as a crucial link connecting the stages of economic production and environmental governance. Mitigating inefficiencies in emissions is instrumental in optimizing resource utilization, mitigating pollution, preserving ecological equilibrium and fostering sustainable economic development. The investigation and enhancement of emission efficiency play a pivotal role in achieving a harmonious balance between economic progress and environmental conservation, thus contributing to the establishment of a more sustainable ecological governance system. Consequently, this section focuses on the analysis of emission inefficiencies.

Figure 7 shows the average provincial MI, RHI and THI for each year and for each region. From Figure 7a, we find that MI and RHI continue to oscillate. MI weakens in 2017, recovers in 2018, declines in 2019 and reaches the maximum in 2020. RHI increases from 2016 to 2017, drops in 2018, increases in 2019 and decreases in 2020. This indicates that MI and RHI have opposite change trends. THI drops significantly from 2016 to 2020. It can be seen that THI has little effect on TEI in 2019 and 2020. Under the combined effect of the three inefficiency indexes, TEI does not improve over time. Compared to the average provincial TEI in 2016, it is downward in 2020, even with the significant increase in RHI or MI in some years. It can be seen from Figure 7b that the primary source of the TEI of China is MI, followed by RHI, and THI is the smallest. In the eastern region, MI accounts for the largest proportion of TEI, followed by THI and the smallest proportion is RHI. For both central and western regions, RHI accounts for the largest proportion, followed by MI, and THI accounts for the smallest. Especially for the central region, the emissions inefficiency mainly derives from the poor regional technology.

Due to the space limitation, Figure 8 shows the TEI and its decomposition from 2016 to 2020 just for several typical provinces. Among the 30 provinces, only four provinces, Beijing, Shanghai, Hainan and Qinghai, have RHI and MI equal to zero in each sample year. This means that the inefficiency of the four provinces only derives from the behindhand emission reduction technology in the corresponding year. The TEI of Shanxi, Inner Mongolia and Shandong is the largest among the 30 provinces in one sample year. In fact, the three provinces are extremely inefficient in terms of emissions each year. They do not have the same inefficiency sources. Shandong's TEI mainly derives from its own low management efficiency; Shanxi's TEI mainly derives from low RHI, except for 2016; and Inner Mongolia's TEI is mainly due to low management efficiency, except for 2019. In 2020, only Anhui's THI is greater than 0 among the 30 provinces, indicating that the emission reduction technology in Anhui is relatively bad in 2020.

3.3.2. TGR Analysis

In order to highlight the advantages of our segmented projection method, Table 9 gives the RTGR and TTGR results calculated with the direct projection method, that is, by using Kao [45]'s NSBM model directly. It can be seen that the TGR results of many provinces are greater than 1, which has been marked in bold in Table 9. This violates the basic property of meta-frontier analysis. Table 10 shows the results of RTGR and TTGR generated by our segmented projection method. Obviously, all the values of of RTGR and TTGR in Table 10 are not greater than unity. Therefore, our method successfully overcomes the issue of TGR > 1.

Figure 9 shows RTGR and TTGR of each region and the whole of China from 2016 to 2020. From Figure 9a, we find that the RTGR in the eastern region always has a high level from 2016 to 2020. In 2018, only the RTGR of Tianjin is 0.83, but the RTGR of other eastern provinces is equal to 1 from 2016 to 2020. This means that for most eastern provinces, there is no gap between the regional frontier and the temporal frontier. With the exception of a slight decline in 2018, TTGR in the eastern region generally shows an upward trend. This means that for the eastern provinces, technology development has gradually reduced the temporal gaps in emission technology. We know from Table 10 that all the eastern provinces

have a TTGR less than one in 2016 and 2018. In 2020, there is no regional technology gap and no temporal technology gap for all the eastern provinces.

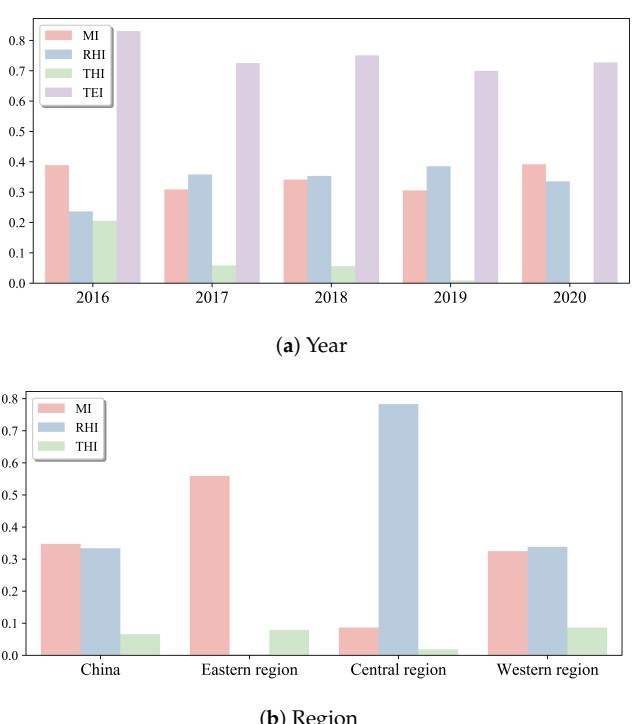

**Figure 7.** Average TEI and its decomposition.

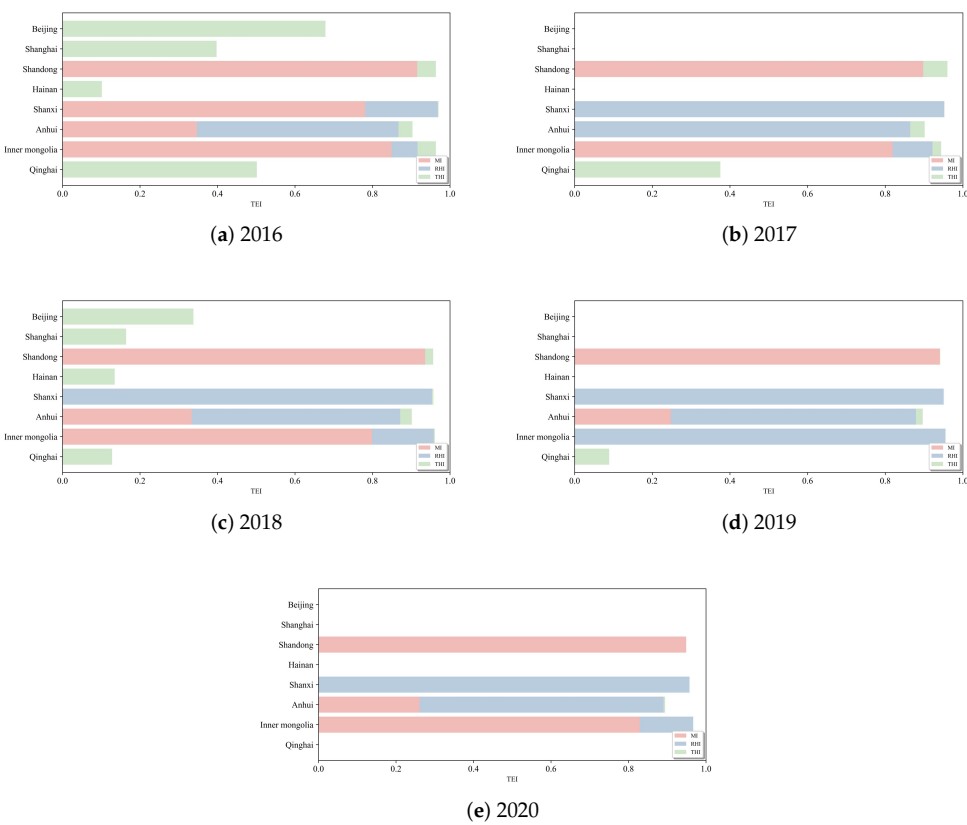

**Figure 8.** Provincial TEI and its decomposition from 2016 to 2020.

**Table 9.** Provincial RTGR and TTGR generated by direct projection.

| Province | 2016 | | 2017 | | 2018 | | 2019 | | 2020 | |
|---|---|---|---|---|---|---|---|---|---|---|
| | RTGR | TTGR | RTGR | TTGR | RTGR | TTGR | RTGR | TTGR | RTGR | TTGR |
| Beijing | 1.00 | 0.32 | 1.00 | 1.00 | 1.00 | 0.66 | 1.00 | 1.00 | 1.00 | 1.00 |
| Tianjin | 1.00 | 0.19 | 1.00 | 1.00 | 0.82 | 0.30 | 1.00 | 1.00 | 1.00 | 1.00 |
| Hebei | 1.00 | 0.56 | 0.88 | 0.62 | 0.94 | 0.76 | **1.02** | 0.87 | **1.11** | 0.97 |
| Liaoning | 0.92 | 0.70 | 0.94 | 0.76 | 1.00 | 0.86 | 1.00 | 0.87 | 1.00 | 1.00 |
| Shanghai | 1.00 | 0.60 | 1.00 | 1.00 | 1.00 | 0.84 | 1.00 | 1.00 | 1.00 | 1.00 |
| Jiangsu | 1.00 | 0.32 | 0.89 | 0.37 | 0.98 | 0.55 | **2.15** | 0.49 | 1.00 | 1.00 |
| Zhejiang | 1.00 | 0.32 | 0.89 | 0.36 | 0.96 | 0.49 | 1.00 | 0.83 | 1.00 | 1.00 |
| Fujian | 1.00 | 0.54 | 0.88 | 0.54 | **1.29** | 0.65 | **1.03** | 0.84 | **1.08** | **1.03** |
| Shandong | 1.00 | 0.32 | 0.89 | 0.36 | 0.99 | 0.62 | 0.99 | 0.82 | 1.00 | 1.00 |
| Guangdong | 1.00 | 0.36 | 0.89 | 0.37 | 0.98 | 0.59 | 1.00 | 0.78 | 1.00 | 1.00 |
| Hainan | 1.00 | 0.90 | 1.00 | 1.00 | 1.00 | 0.87 | 1.00 | 1.00 | 1.00 | 1.00 |
| Shanxi | 0.15 | 0.74 | 0.05 | 0.82 | 0.05 | 0.86 | 0.05 | 0.71 | 0.04 | 1.00 |
| Jilin | 0.16 | 0.71 | 0.14 | 0.82 | 0.18 | 0.86 | 0.16 | 0.87 | 0.16 | 1.00 |
| Helongjiang | 0.15 | 0.73 | 0.09 | 0.82 | 0.18 | 0.86 | 0.11 | 0.87 | 0.16 | 1.00 |
| Anhui | 0.20 | 0.43 | 0.14 | 0.49 | 0.18 | 0.74 | 0.14 | **1.70** | 0.13 | **1.02** |
| Jiangxi | 0.14 | 0.73 | 0.14 | 0.76 | 0.15 | 0.86 | 0.15 | 0.87 | 0.16 | 1.00 |
| Henan | 0.20 | 0.40 | 0.18 | 0.36 | 0.12 | 0.66 | 0.13 | 0.85 | 0.11 | 1.00 |
| Hubei | 0.20 | 0.42 | 0.20 | 0.49 | 0.17 | 0.79 | 0.15 | 0.74 | 0.12 | 1.00 |
| Hunan | 0.18 | 0.42 | 0.18 | 0.49 | 0.16 | 0.67 | 0.16 | 0.91 | 0.14 | **1.14** |
| Inner Mongolia | 0.33 | 0.67 | 0.37 | 0.57 | 0.24 | 0.86 | 0.04 | 0.87 | 0.25 | 1.00 |
| Guangxi | 0.55 | 0.24 | 0.27 | 0.82 | 0.17 | 0.86 | 0.33 | 0.87 | 0.32 | 1.00 |
| Chongqing | 1.00 | 0.21 | 0.24 | 0.71 | 0.25 | 0.86 | 0.27 | 0.87 | 0.26 | 1.00 |
| Sichuan | 0.21 | 0.42 | 0.23 | 0.49 | 0.21 | 0.77 | 0.18 | 0.94 | 0.16 | 1.00 |
| Guizhou | 0.24 | 0.71 | 0.39 | 0.82 | 0.41 | 0.86 | 0.41 | 0.87 | 0.40 | 1.00 |
| Yunnan | 0.32 | 0.74 | 0.30 | 0.82 | 0.30 | 0.86 | 0.31 | 0.87 | 0.30 | 1.00 |
| Shaanxi | 0.26 | 0.43 | 1.00 | 0.65 | 1.00 | 0.93 | 1.00 | 1.00 | 1.00 | 1.00 |
| Gansu | 0.53 | 0.70 | 0.49 | 0.82 | 0.54 | 0.86 | 0.55 | 0.87 | 0.50 | 1.00 |
| Qinghai | 1.00 | 0.49 | 1.00 | 0.62 | 1.00 | 0.87 | 1.00 | 0.91 | 1.00 | 1.00 |
| Ningxia | 0.53 | 0.22 | 0.11 | 0.87 | 0.15 | 0.76 | 0.14 | 0.79 | 0.20 | 0.99 |
| Xinjiang | 0.38 | 0.72 | 0.20 | 0.82 | 0.22 | 0.73 | 0.17 | 1.17 | 0.17 | 1.00 |

It can be seen from Figure 9b that the RTGR in the central region is very low and shows a downward trend as a whole. This means that in the central region, the gap between the regional frontier and the temporal frontier is large and is widening. From Table 10, we see that in the sample period, all the central provinces obtain a small RTGR. In contrast, the TTGR in the central region increases steadily year-by-year. In 2020, only one central province has a TTGR less than 1, and the rest are equal to 1. The gap between the temporal frontier and the meta-frontier is shrinking in the central region.

As can be seen from Figure 9c, 2016 is a special year for the western region. In this year, the RTGR is larger than the TTGR. From Table 10, the RTGR values of Chongqing, Guizhou, Shaanxi and Qinghai in 2016 are equal to one. This may be due to the positive regional results brought by the development of China's western regions and the advancement of the "Belt and Road" construction. Obviously, the RTGR in the western region is higher than that in the central region, but it also shows a decreasing trend generally. This indicates that the gap between the regional frontier and the temporal frontier in the western region is smaller than that in the central region, but it is also widening. The TTGR in the western region shows an increasing trend year-by-year. In 2020, the TTGR of all western provinces is equal to 1, which means that there is no gap between the temporal frontier and the meta-frontier.

For the whole of China (see Figure 9d), RTGR has continued to decline since 2016, indicating that the gap between the regional frontier and the temporal frontier is increasing. Conversely, TTGR generally rises over the sample period. Namely, the gap between the temporal frontier and the meta-frontier shrinks from 2016 to 2020 in general. China's emissions inefficiency is jointly affected by both regional and temporal heterogeneities, and RTGR is bigger than TTGR in 2016. With the development of technology from 2017 to 2020, RTGR is smaller than TTGR. So, the two kinds of heterogeneity have different effects on TGR. To a certain extent, this characteristic verifies the rationality of considering regional and temporal heterogeneities in this paper.

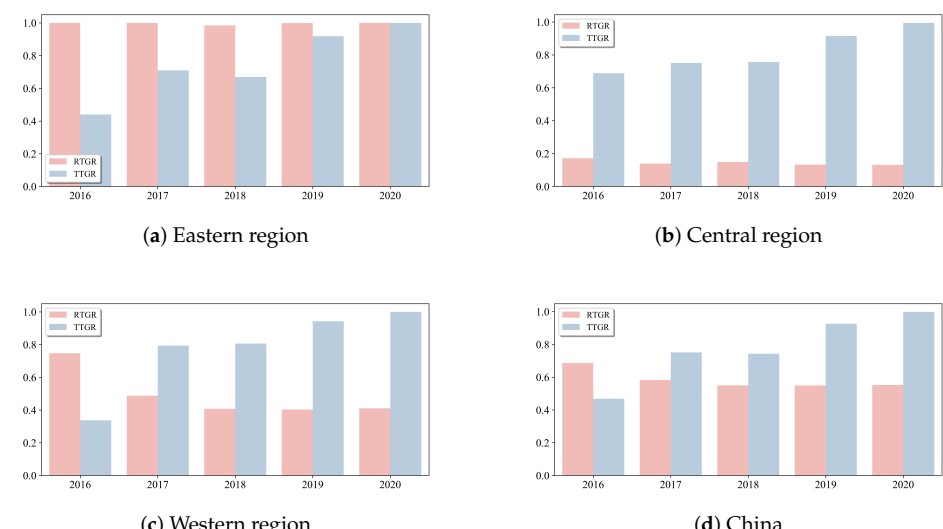

**Figure 9.** RTGR and TTGR from 2016 to 2020.

**Table 10.** Provincial RTGR and TTGR generated by our segmented projection method.

| Province | 2016 | | 2017 | | 2018 | | 2019 | | 2020 | |
|---|---|---|---|---|---|---|---|---|---|---|
| | RTGR | TTGR | RTGR | TTGR | RTGR | TTGR | RTGR | TTGR | RTGR | TTGR |
| Beijing | 1.00 | 0.32 | 1.00 | 1.00 | 1.00 | 0.66 | 1.00 | 1.00 | 1.00 | 1.00 |
| Tianjin | 1.00 | 0.19 | 1.00 | 1.00 | 0.83 | 0.65 | 1.00 | 1.00 | 1.00 | 1.00 |
| Hebei | 1.00 | 0.56 | 1.00 | 0.54 | 1.00 | 0.66 | 1.00 | 0.70 | 1.00 | 1.00 |
| Liaoning | 1.00 | 0.44 | 1.00 | 0.71 | 1.00 | 0.86 | 1.00 | 1.00 | 1.00 | 1.00 |
| Shanghai | 1.00 | 0.60 | 1.00 | 1.00 | 1.00 | 0.84 | 1.00 | 1.00 | 1.00 | 1.00 |
| Jiangsu | 1.00 | 0.32 | 1.00 | 0.64 | 1.00 | 0.54 | 1.00 | 0.88 | 1.00 | 1.00 |
| Zhejiang | 1.00 | 0.32 | 1.00 | 0.32 | 1.00 | 0.47 | 1.00 | 0.97 | 1.00 | 1.00 |
| Fujian | 1.00 | 0.54 | 1.00 | 0.46 | 1.00 | 0.62 | 1.00 | 0.74 | 1.00 | 1.00 |
| Shandong | 1.00 | 0.32 | 1.00 | 0.32 | 1.00 | 0.62 | 1.00 | 0.96 | 1.00 | 1.00 |
| Guangdong | 1.00 | 0.32 | 1.00 | 0.81 | 1.00 | 0.58 | 1.00 | 0.87 | 1.00 | 1.00 |
| Hainan | 1.00 | 0.90 | 1.00 | 1.00 | 1.00 | 0.87 | 1.00 | 1.00 | 1.00 | 1.00 |
| **Eastern** | **1.00** | **0.44** | **1.00** | **0.71** | **0.98** | **0.67** | **1.00** | **0.92** | **1.00** | **1.00** |
| Shanxi | 0.15 | 0.90 | 0.05 | 1.00 | 0.05 | 0.87 | 0.05 | 1.00 | 0.04 | 1.00 |
| Jilin | 0.16 | 0.82 | 0.14 | 1.00 | 0.18 | 0.87 | 0.16 | 1.00 | 0.16 | 1.00 |
| Heilongjiang | 0.15 | 0.87 | 0.09 | 1.00 | 0.18 | 0.87 | 0.11 | 1.00 | 0.16 | 1.00 |
| Anhui | 0.20 | 0.60 | 0.14 | 0.64 | 0.19 | 0.74 | 0.17 | 0.67 | 0.15 | 0.96 |
| Jiangxi | 0.14 | 0.90 | 0.14 | 1.00 | 0.15 | 0.87 | 0.15 | 1.00 | 0.16 | 1.00 |
| Henan | 0.20 | 0.40 | 0.18 | 0.38 | 0.12 | 0.66 | 0.13 | 1.00 | 0.11 | 1.00 |
| Hubei | 0.20 | 0.49 | 0.20 | 0.46 | 0.17 | 0.62 | 0.14 | 1.00 | 0.13 | 1.00 |
| Hunan | 0.18 | 0.52 | 0.18 | 0.53 | 0.16 | 0.58 | 0.16 | 0.66 | 0.14 | 1.00 |
| **Central** | **0.17** | **0.69** | **0.14** | **0.75** | **0.15** | **0.76** | **0.13** | **0.92** | **0.13** | **0.99** |
| Inner Mongolia | 0.61 | 0.36 | 0.48 | 0.72 | 0.22 | 0.87 | 0.04 | 1.00 | 0.20 | 1.00 |
| Guangxi | 0.55 | 0.24 | 0.30 | 1.00 | 0.17 | 0.87 | 0.33 | 1.00 | 0.33 | 1.00 |
| Chongqing | 1.00 | 0.21 | 0.25 | 1.00 | 0.25 | 0.87 | 0.27 | 1.00 | 0.29 | 1.00 |
| Sichuan | 0.21 | 0.49 | 0.23 | 0.45 | 0.22 | 0.51 | 0.19 | 0.67 | 0.16 | 1.00 |
| Guizhou | 1.00 | 0.21 | 0.59 | 0.67 | 0.41 | 0.87 | 0.41 | 1.00 | 0.40 | 1.00 |
| Yunnan | 0.62 | 0.38 | 0.31 | 0.98 | 0.30 | 0.87 | 0.32 | 1.00 | 0.30 | 1.00 |
| Shaanxi | 1.00 | 0.21 | 1.00 | 0.65 | 1.00 | 0.93 | 1.00 | 1.00 | 1.00 | 1.00 |
| Gansu | 0.98 | 0.48 | 0.90 | 0.63 | 0.54 | 0.87 | 0.55 | 1.00 | 0.50 | 1.00 |
| Qinghai | 1.00 | 0.50 | 1.00 | 0.62 | 1.00 | 0.87 | 1.00 | 0.91 | 1.00 | 1.00 |
| Ningxia | 0.53 | 0.21 | 0.11 | 1.00 | 0.15 | 0.77 | 0.14 | 0.79 | 0.20 | 1.00 |
| Xinjiang | 0.72 | 0.41 | 0.20 | 1.00 | 0.22 | 0.61 | 0.18 | 1.00 | 0.13 | 1.00 |
| **Western** | **0.75** | **0.34** | **0.49** | **0.79** | **0.41** | **0.81** | **0.40** | **0.94** | **0.41** | **1.00** |
| **China** | **0.69** | **0.47** | **0.58** | **0.75** | **0.55** | **0.74** | **0.55** | **0.93** | **0.55** | **1.00** |

## 4. Discussion and Suggestions

### 4.1. Discussion

Building upon the aforementioned empirical findings, we turn our attention to the discussion of eco-efficiency within each category, coupled with an exploration of emission inefficiency and TGR.

(1)  In the realm of management eco-efficiency, several noteworthy observations emerge:

- Over the majority of sample years, the managerial performance in economic production across most Chinese provinces surpasses that in environmental governance.
- Contrary to the conventional pattern where the eastern region outperforms the central and western regions, the central provinces exhibit commendable management eco-efficiency, while the eastern provinces lag behind.
- Throughout China, the overall management technical level remains relatively stable across the sample period, except for 2020, when a discernible change occurs.

(2)  In the context of regional eco-efficiency, additional insights surface:

- In the majority of provinces and sample years, economic production influenced by regional technologies outperforms environmental governance determined by the same technologies.
- In Central and Western China, regional eco-efficiency, as determined by regional technologies, shows a declining trend year-by-year, eventually stabilizing. This trend signifies a widening and stabilizing gap in eco-efficiency dependent on regional technologies between the central and western regions versus the eastern regions.
- The technological advancements in economic production and environmental governance in the eastern provinces significantly outpace those in the central and western provinces, resulting in excellent regional eco-efficiency in the eastern region and a comparative disadvantage for the central region.

(3)  In the domain of temporal eco-efficiency, the following observations can be made:

- For the majority of provinces in most years, TPE surpasses TGE, akin to the patterns observed in MPE versus MGE, as well as RPE versus RGE. This consistent pattern indicates that, in most provinces and years, their management level, regional production technology and temporal production technology are more conducive to economic production than environmental governance.
- Temporal efficiency indexes generally exhibit a positive trend, indicating the increasing role of scientific and technological development in promoting economic production and environmental governance. Moreover, this trend is not specific to any one region, as there is no significant regional difference in the promoting effect of scientific and technological development on economic production and environmental governance.

(4)  Regarding emissions inefficiency, the following discoveries come to light:

- The emissions efficiency of most provinces has gradually improved, despite some fluctuations in the TEI in intervening years.
- The ongoing development of technology in economic production and environmental management in China weakens emissions inefficiency dependent on temporal technology.
- Poor management levels represent the primary obstacle to improving emissions efficiency in the eastern region, while the level of regional technology is the major factor contributing to emissions inefficiency in the central and western regions. For the western region, poor management also plays a significant role in emissions inefficiency. To enhance national emissions efficiency, concerted efforts are required to improve both management and regional technology levels.

(5)  Turning to the results of TGR, several key observations are made:

- Similar to the eco-efficiency indexes, the eastern provinces exhibit clear regional advantages in emissions efficiency, with emission reduction technology showing continuous development. This underscores that the superior emissions efficiency in the eastern provinces results from a combination of regional and technological advancements.

- While the development of emission reduction technologies aids the central and western provinces in reducing emissions inefficiency, their lower RTGR emphasizes the importance of learning advanced emission reduction technologies from the eastern provinces to narrow the regional technology gap and prevent further widening.
- The role of national technology in promoting emissions efficiency increases steadily over the years. Macro policies, such as the "Opinions of the Central Committee of the Communist Party of China and the State Council on Accelerating the Construction of Ecological Civilization" in April 2015, act as accelerators for improving TTGR. Additionally, attention should be paid to the widening gap in emissions efficiency caused by regional differences.

*4.2. Suggestions*

Building on the preceding discussions, the following suggestions are proposed:

(1) Enhancing eco-efficiency necessitates a balanced consideration of both economic production and environmental governance. Rather than solely pursuing economic development at the expense of environmental concerns, it is crucial for the government to take a proactive stance in augmenting its capabilities in environmental governance. This involves increased investments in environmental management, reinforced supervision of local industrial enterprises and concerted efforts to address pollution at its source. Such measures aim to realize high-quality and sustainable development.

(2) To enhance eco-efficiency and mitigate emissions, the eastern region can concentrate on elevating its own management proficiency by drawing insights from the successful management experiences of the central provinces. Simultaneously, the central region should focus on narrowing the regional technology gap through intensified technical exchanges with the eastern region and the adoption of advanced technologies and valuable resources. The western region should navigate a balanced approach, addressing both management enhancement and regional technology considerations. At the national level, government policies should be introduced to incentivize provinces in promoting effective management practices and reducing regional technology disparities.

## 5. Conclusions, Limitations and Future Work

The existing studies do not consider stage division as well as both regional and temporal technology heterogeneity on the premise of ensuring that TGRs are not greater than 1. To fill the gaps of existing research, the following work was carried out in this study:

Theoretically, we introduced a three-level meta-frontier NSBM approach for a two-stage network structure. We first considered both regional and temporal technology heterogeneities to build three kinds of frontiers: regional frontiers, temporal frontiers and the meta-frontier. Furthermore, the activities of the DMUs were divided into the EP stage and the EG stage. The intermediate variables connecting the two stages are the pollutant emissions. Based on these, a three-level meta-frontier NSBM approach was built, and nine efficiency indexes are proposed. In order to deeply analyze the emissions efficiency, we defined the indexes of MI, RHI as well as THI and constructed the TEI index. In addition, in order to analyze the technology gaps, we constructed RTGR and TTGR indexes.

In application, we used the proposed three-level meta-frontier NSBM model to evaluate the eco-efficiency of 30 Chinese provinces during the 13th Five-Year Plan period. The empirical results demonstrate that the economic production performance is better than the environmental governance performance for most provinces in most years. Large gaps exist in regional and temporal eco-efficiencies. The central region has the best management eco-efficiency, followed by the western region and the eastern region; the eastern region has the best regional eco-efficiency, followed by the western region and the central region; and the temporal eco-efficiencies of the three regions generally increase year-by-year. Based on the TEI and its decomposition inefficiency indexes, it can be seen that MI accounts for the largest proportion of the TEI of China, followed by RHI, and THI accounts for the

smallest. In addition, for the entirety of China, the gap between the regional frontier and the temporal frontier has widened since 2016, and the gap between the temporal frontier and the meta-frontier generally shrinks from 2016 to 2020.

Our study takes a comprehensive approach, considering region, time, stage and network structure. By dividing ecological activities into economic production and environmental governance stages, we gain a detailed understanding of ecological efficiency dynamics. The incorporation of emission efficiency analysis and the application of a two-stage three-level meta-frontier network model address the challenge of TGR exceeding 1. This multifaceted research perspective enhances insights into the critical factors influencing ecological and emission efficiency. However, our study has limitations.

(1) The availability of data restricted the length of our sample period, limiting the scope of our analysis.
(2) Too many inputs and outputs will reduce the efficiency discrimination ability of DEA models. In order to maintain the efficiency discrimination ability, the proposed three-level meta-frontier NSBM approach only takes the emissions of $SO_2$ and $CO_2$ as intermediate variables and does not consider other pollutants.
(3) The three-level meta-frontier NDEA model proposed in this paper only considers the regional and temporal technology heterogeneities.

In future research work, we can carry out further research involving the following aspects:

(1) In order to discriminate the eco-efficiency and meanwhile to consider more pollutant emissions, such as waste gas, waste water and solid waste, the  three-level meta-frontier network cross-efficiency [49,50] or the super-efficiency [51] approach can be built.
(2) We can consider other heterogeneities according to the actual situation, such as industry categories and scales, to introduce more production frontiers and propose the corresponding multi-level meta-frontier NDEA models.
(3) The Tobit regression model [52] is one of the commonly used methods for discussing the external factors that affect efficiency. Due to the space limitation, we did not use the Tobit regression here. To analyze the significance of external influencing factors for the emissions inefficiency or eco-efficiency scores, we can select appropriate explanatory variables for Tobit regression analysis.

**Author Contributions:** Conceptualization, R.L. and X.W.; Formal analysis, R.L.; Investigation, R.L. and Y.J.; Methodology, R.L. and X.W.; Data collection, Y.J.; Project administration, R.L.; Software, X.W.; Validation, R.L.; Writing-original draft, R.L. and X.W. All authors have read and agreed to the published version of the manuscript.

**Funding:** This research was funded by the National Natural Science Foundation of China [grant number 71971163].

**Data Availability Statement:**  The datasets used and/or analyzed during the current study are available from the corresponding author on reasonable request.

**Conflicts of Interest:** The authors declare no conflicts of interest. The funders had no role in the design of the study; in the collection, analyses, or interpretation of data; in the writing of the manuscript; or in the decision to publish the results.

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
