# Peer review of "Ecological Efficiency Measurement and Technical Heterogeneity Analysis in China: A Two-Stage Three-Level Meta-Frontier Network Model Based on Segmented Projection"

_systems, doi:10.3390/systems12010022_

Round 1
Reviewer 1 Report
Comments and Suggestions for Authors
This article proposes a two-stage three-level meta-frontier model by combining dual heterogeneity and stage divisions, and applies the proposed model to analyze the eco-efficiencies of 30 provinces in China. The emissions inefficiencies and its decomposition indicators are also analyzed. The empirical analysis in this paper is very detailed, and the correlation figures and tables are also abundant. It is very creative to combine the two-stage network SBM model with the three-level meta frontier model. The detailed comments are as follows.
(i) The authors pay special attention to the inefficiency of pollution emission. It is not only decomposed into three inefficiency indicators, but also the corresponding technology gap ratios is discussed. So I think it is necessary to give some reasons and significance for the analysis of emissions inefficiency in section 3.3.
(ii) I feel that the overall length of the article is a little long. I suggest that the authors delete some descriptions in Section 3.
(iii) In section 3, it should be “Mann Whitney U test”, not “Mann Whitney test” . Mann Whitney U test is to judge the difference between the average values of the two groups of data. If the p-value is less than the significance level (5% here), it is proved that there is a significant difference between the two groups of data; otherwise, there is no difference between the two groups of data statistically. As for the comparison, it was determined by Mann Whitney’s null hypothesis and alternative hypothesis. Here, the null hypothesis should be H0: µ1>=µ2, and the alternative hypothesis H1: µ1 < µ2. If the p-value obtained is less than the significance level (5%), the null hypothesis is rejected. I think it would be better for the authors to add descriptions of the null hypothesis and alternative hypothesis of the Mann Whitney U test.
(iv) Sections 3.3 and 3.4 are suggested to be merged into one section, because both of them are related to emissions inefficiency.
(v) There are other literature about the issue of TGR >1. Please cite them.
[1]Wang, Q., Hang, Y., Hu, J.L., Chiu, C.R., 2018. An alternative meta-frontier framework for measuring the heterogeneity of technology. Naval Research Logistics (NRL) 65, 427–445.
[2]Chen, Y., Xu, W., Zhou, Q., Zhou, Z., 2020b. Total factor energy efficiency, carbon emission efficiency, and technology gap: Evidence from sub-industries of Anhui province in china. Sustainability 12.
[3]Chen, L., Huang, Y., Li, M.J., Wang, Y.M., 2020a. Meta-frontier analysis using cross-efficiency method for performance evaluation. European Journal of Operational Research 280, 219-229.
[4]Lin, R., Peng, Y., 2023. A new cross-efficiency meta-frontier analysis method with good ability to identify technology gaps. European Journal of Operational Research, online published.
[5]Yu, M.M., Rakshit, I., 2023. Assessing the dynamic efficiency and technology gap of airports under different ownerships: A union dynamic NDEA approach. Omega 119, 102888.
[6]Yu, M.M., See, K.F., Hsiao, B., 2022. Integrating group frontier and meta-frontier directional distance functions to evaluate the efficiency of production units. European Journal of Operational Research 301, 254-276.
Reviewer 2 Report
Comments and Suggestions for Authors
This paper addresses an important and interesting problem- the ecological efficiency measurement and technical heterogeneity analysis in China. However, many issues still need to be improved.
1.The abstract needs to be rewritten, and the main research conclusions have not been reflected
2.The introduction mainly discusses research methods, which can be expanded to discuss indicator systems, research perspectives, and other aspects.
3.The introduction of research methods is too much and lacks an explanation of indicator systems.
4.It is best to use ArcGIS to display spatial differences
5.The discussion is too simplistic. What are the contributions of this paper? Meanwhile, the shortcomings of this article have not been identified.
Comments on the Quality of English LanguageThere are many typos and grammar errors in this paper.
Reviewer 3 Report
Comments and Suggestions for Authors
Please see the attached file.

Round 2
Reviewer 2 Report
Comments and Suggestions for Authors
I think it can be published now